# The Eco-Cathedric City: Rethinking the Human–Nature Relation in Urbanism

**Rob Roggema** 

Tecnologico de Monterrey, Escuela de Arquitectura, Arte y Diseño, Campus Monterrey, Monterrey 64849, Mexico; rob.roggema@tec.mx

**Abstract:** Current planning of urban landscapes is dominated by a human-centric view. This leads to short-term orientation, predictable planning outcomes, and decisions being taken by a small group of humans. Alternatively, a symbiotic human–nature relationship could be a prelude to a balanced future in which sustaining all living organisms prevails. In this article, a novel approach to designing such an urban landscape is presented: the Eco-cathedric City. In this proposition, the design process thrives on high complexity, deep uncertainty, contingent nature–human relations, slow urbanism, and imaginability. It is concluded that three mechanisms should be core to this approach: (eco-)cathedral thinking, considering the impact of current decisions on seven future generations; (eco-)acupuncturist design, which plans for the process by igniting a single small intervention; and (eco-)cracy, in which a variety of actors, human and non-human organisms, co-decide. In a practical sense, the Eco-cathedric City finds its foundation in understanding local ecosystems and using this knowledge to design a self-organizing ecosystem in which regenerative resource management is prioritized, after which social constructs are formed to support this design and to fit human uses within the boundaries of this framework to conclude with an evolving belief system in which reciprocity and symbiocity are the core values.

**Keywords:** Eco-cathedric City; eco-cracy; eco-acupuncture; urban landscape; Anthropocene

## 1. Introduction

The percentage of the world's population that lives in urban areas is expected to increase to 68% by 2050, according to projections of the UN [1]. Due to their large number, the people living in cities have a profound impact on their environment and the use of natural resources. Urbanization has a direct and indirect impact on biodiversity [2]. The expansion of urban areas directly leads to physical alterations, such as modified soils and altered disturbances, habitat loss, or degradation. Indirectly, biodiversity is impacted by altered water and nutrient availability, air pollution, increased competition from non-native species, and altered herbivory and predation rates [3].

Additionally, urbanization is both a cause and a victim of climate change. Cities account for 70–75% of carbon emissions [4,5]. Human activities, such as the use of fossil fuels, land-use change, and agricultural practice (all directly or indirectly related to urban life), increase concentrations of greenhouse gases, leading to higher temperatures and changes in precipitation, soil moisture, and sea level. At the same time, an increased population density, especially in sensitive areas where often much urbanization occurs (such as along coasts and in river basins), places many more people in a vulnerable position [6]. Rapid urbanization increases human vulnerability even more [7]. Human influence on the planet has been so profound that a new geological epoch has been declared: 'considering major and still growing impacts of human activities on earth and atmosphere, and at all, including global, scales, it is more than appropriate to emphasize the central role of mankind in geology and ecology by using the term "Anthropocene" for the current geological epoch' [8,9].

These changes place the planet at risk of crossing several planetary boundaries [10]. In the last decade, the number of aspects of the planetary system that have crossed their boundaries has steadily increased [11,12] (Figure 1).

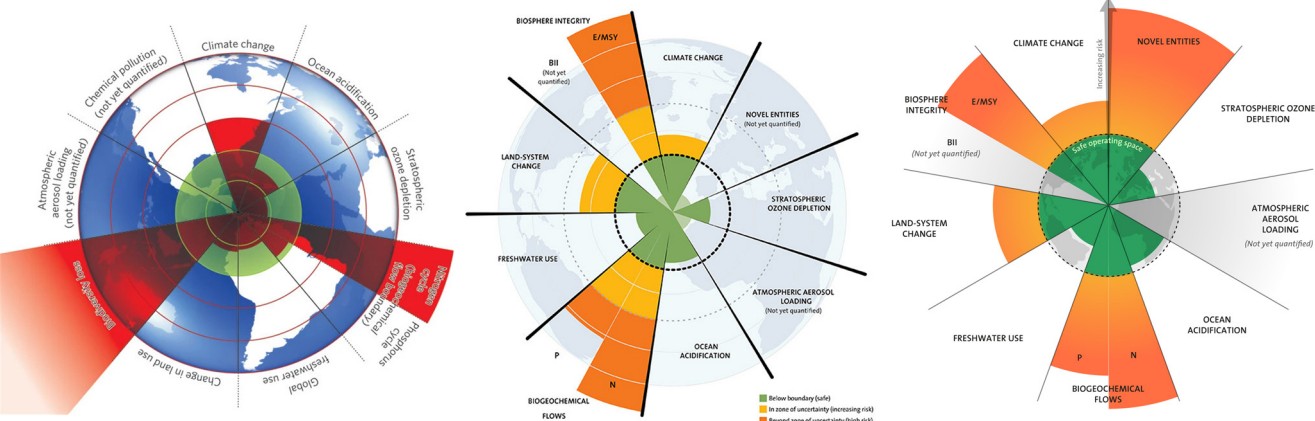

**Figure 1.** Planetary boundaries as estimated in 2009, 2015, and 2022 (credits: [10]; J. Lokrantz/Azote based on [12]; Azote for Stockholm Resilience Centre, based on analysis in [11,12]).

The role that urbanization plays in this mechanism of increasing pressure on the planetary system requires a fundamental rethink of the way cities are planned, are developed, and develop. Current drivers for urbanization vary significantly regionally [13]. Factors such as the annual GDP, population growth, international capital flows, informal economy, land-use policies, and transport costs all are major factors that play out very different locally [14]. Most of these factors, however, are not directly driven by care for the planetary system. It is more likely that they form a toxic mix that drives planetary change across Earth's safe boundaries.

This means that for urbanization, other drivers should play a more prominent role and novel spatial concepts are needed for how cities grow to stay within a safe operating space. It is not very likely that urban planning alone will get humanity to operate within its planetary boundaries; however, the invention of novel urbanization models can be useful in transforming urban impact from burden to benefit.

To enhance thinking about such a future perspective, three pillars are further explored in this article: long-term thinking about eco-cathedrals [15] and the idea that human–nature is one cosmology [16], leading us into the symbiocene [17]. These foundations shape up thinking towards a new urban development model: the Eco-cathedric City.

## 2. Materials and Methods

The investigation consists of several building blocks: a literature review, conceptualizing foundations, and construction of a pathway for future urbanization (Figure 2). Firstly, the literature review summarizes the major impacts of current urbanization and the range of responses that urban planners and academics have developed. The incompatibility of impact and response has led to a third block of literature, investigating the deeper causes of the mismatch and ignorance towards or even neglect of inconvenient confrontations with reality, such as uncertainties and complexity. Literature studies have subsequently fed a creative process of defining the counteractive foundations for future urban development. In a series of loosely structured interviews and conversations with thought leaders (7) in urbanism from around the world, major components (long-term futures and human–nature relations) have been derived and loaded with content. The final step in the research has been the conceptualization of a pathway for future urbanism, driven by these foundations. What-if questions have been the major factor in three interactive, online working sessions. What if, in the long-term, the natural environment and a symbiotic future drive urban development; how would we design, plan, and decide about the cities we live in? Outcomes

have been synthesized in a conceptual model of the Eco-cathedric City, consisting of three focus areas: time, space, and decision.

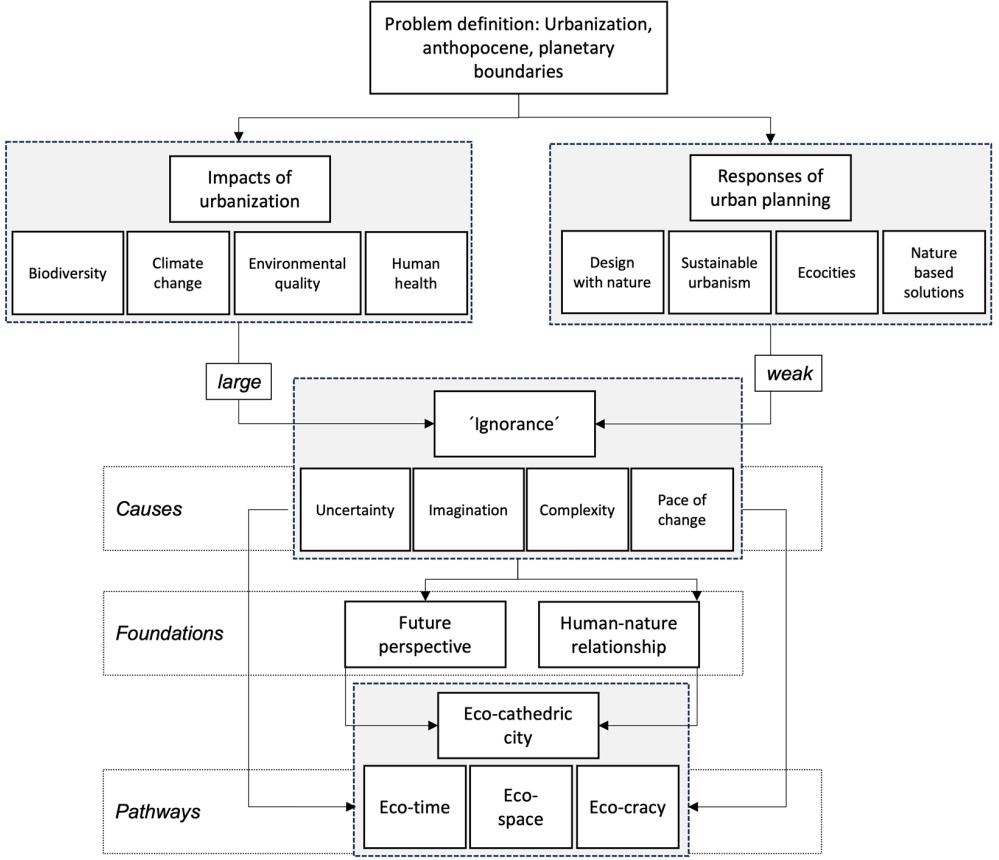

**Figure 2.** Methodology and lines of research.

## 3. Literature Review

### 3.1. The Impact of Cities

Many urbanization processes continue to impact planetary conditions, contributing to the crossing of its boundaries.

When cities grow, biodiversity is impacted both directly and indirectly [2] as well as physically and environmentally [3]. Globally, 69% of biodiversity has been lost since 1970 [18,19], land-use change being the most profound cause [20]. For instance, bird population decline is driven by monofunctional agriculture, providing for urban food demand [21].

Cities account for 71–76% of $CO_2$ emissions [22], as urban activities are seen as the major source of these emissions [23]. Ironically, this way, urbanization contributes to its own rising surface temperatures [24] with an estimated 0.05 °C per decade [25].

Vegetation and green are impacted by urbanization in combination with climate change [26]. Urban green spaces change [27] as they are occupied [28] and fragmented [29]. Consequently, urban temperatures rise [30,31], which decreases quality of life [32,33]. In Accra, for instance, urban land use has grown from 55.1% to 83.79% while green spaces have reduced from 41% to 15% in 27 years [34].

Urbanization impacts the environment, causing problems with regards to air, water, and soil.

In urban environments, $CO_2$ and airborne pollutants affect approximately 50% of the global population [35–37], making them some of the most influential drivers of diseases [38,39]. Only 12% of people living in cities meet the WHO air-quality standards [40].

Urban growth alters natural processes and resource consumption, changes infiltration and evapotranspiration, and exacerbates runoff [41], resulting in an increase in flood risk and reduction in water quality [42]. Increasing summer droughts in combination with severe rainfall and floods cause alien species invasion, enhance toxic algae blooms, and lower oxygen levels, while storms flush nutrients through urban areas or generate acid pulses in the water catchment [43].

Urbanization influences both the amount of contaminated soils [44–47] and the occupation of high-quality soils [48–50], with urban sprawl using more than compact urban growth [51]. Urban diffusion reduces regional biodiversity and food security by land-use change and fragmentation of the landscape [52].

Finally, the city exerts negative impacts on both mental [53] and physical [54] health. Mental health aspects such as loneliness, violence, high crime rates, homelessness, noise and other pollutants, traffic accidents, drug abuse, and insufficiency of mental health services [55] often accumulate in urban areas [56]. Whereas physical health problems in urban areas in developing countries consist of specific infections and western lifestyle diseases, in developed countries physical health problems are of another nature: social breakdown, pollution, accidents and violence, poor nutrition due to poverty, lack of exercise and anxiety, and influx of migrant populations with unequal access to health services and education [54].

Urbanization has profound impacts on the global environment. It impacts biodiversity; urban climate; green spaces; air, water, and soil quantities and qualities; and the mental and physical health of urban dwellers. It asks for a fundamental rethink of how the city is planned.

### 3.2. The Responses of Urban Planning

The problem of urbanization impacting natural conditions has been the subject of responses in urban planning for a long time.

#### 3.2.1. Design with Nature

Ian McHarg was one of the main voices to rethink the way cities are planned. With the ecology and character of the landscape, planning with nature allows cities to become regenerative and grow a stronger sense of place. By including landscape and ecology as a fundamental layer, the fitness of the land for specific uses can be decided [57,58]. This way, nature is used to guide sustainable urbanism [59], responsiveness to human well-being, social improvement, and social hope [60]. It requires the application of sustainability and resilience in design, planning, and administration of the city [61].

#### 3.2.2. Sustainable Urbanism

Historically, sustainable propositions in urban planning have undergone significant changes, from a focus on aesthetics, via rationalization, towards emergent, adaptive ways of planning [62]. Sustainable urbanism features three fundamental qualities: co-creating the city with the citizen as a (design) expert; the city as a (complex) system in which processes of metabolism occur, closing cycles at the lowest possible scale; and taking the ecological landscape as the basis for urban patterns, development, and growth [63]. In designing such a city, redundancy, anti-fragile solutions that grow a stronger city under disruptive change, and nonconventional rule-breaking concepts must be built into designs [62].

#### 3.2.3. Ecocities

Ecocities aim to create a healthy and self-sustaining city [64], modelled on natural ecosystems [65], through integrated urban planning, protecting and nurturing natural assets for the future. It will not compromise future generations, and is governed by the laws of nature, where communities initiate their own solutions; technology is used for self-reliance in harmony with nature, hyperlocal and diverse; and social ecology emerges from cross-cutting committed citizenry [66].

### 3.2.4. Landscape First

Landscape-based regional design [67] shapes the physical form of regions by using the landscape as the basic condition [68] and considers the biosphere to be the context for long-term urban development [69,70]. It envisions a long-term future, identifies common directions for spatial design, and creates inspiring ecologies in a regional-landscape-based plan [71]. It can provide sufficient natural resources by embedding these in the landscape [72] and can bring the hidden ecologies of the landscape in denser conditions back to the surface [73], using historic dynamics of the landscape to regenerate the land [74,75].

### 3.2.5. Nature-Based Solutions

Nature-based solutions are seen as valuable ingredients for sustainable urban planning [76]. By using environmental technology, systems design, urban design, and urban planning [77], a resilient urban future emerges where the city is seen in a systemic way, people and biodiversity benefit, inclusive solutions are generated, context is part of the planning process, and communication and learning are fostered [78].

### 3.2.6. Biourbanism

The concept of biourbanism sees the city as a hyper-complex organism [79] and is seen as a form of nature [80] which strives for optimum efficiency of the system and humans' quality of life [81,82]. It aims for connecting the natural needs of humans and the ecosystem, "deepening the organic interaction between cultural and physical factors in urban reality". Designs that do not follow this produce anti-natural environments, hence failing to enhance life by any means [83]. As an urban planning and design model, biourbanism creates healthier and resilient cities by transforming cities in more equitable, vibrant, climate-resilient places [80,84].

Each of the urban planning responses aims at resetting the urban system as a natural system, thereby limiting climate change and biodiversity loss, and providing a healthy, equal, and prosperous place for humankind. At the same time, a major gap between the ubiquity of these well-meant concepts in the urban resilience discourse and actual implementation is witnessed. Current trends in climate change and biodiversity loss only worsen [85].

## 4. Ignorance of the Future

Although many of responses propose the long term as the foundation for urban planning, at the same time these concerns are ignored in practice (Table 1). There are good reasons for using a long-term perspective [86], which seemingly are not contested in theoretical urban planning responses.

**Table 1.** Ignored concerns and possible ways forward (after [86]).

| Ignored Concern | Possible Alternative Pathway |
|---|---|
| Earth is a resource | Earth is a complex web of the living and non-living |
| Generated signals of the global system (climate, biodiversity) | Listen to the signals and their implications for behavior and policy |
| Understanding of the global system (limits to growth) | Factor in understanding in all major decisions |
| Extremes of growth addiction can be absorbed by the global system without serious damage | Replace by an encompassing and durable ethic |
| Economic rationalism which leads to collapse of the global system | Shift to steady-state economies: the sooner the better |
| Climate denial as a poor substitute for action | Respond to global warming, e.g., through reestablishing civilization on a more enduring basis |
| Peaked over-demand is seen as collapse and descent | Engage seriously with the real prospects of peaked over-demand [87] (Floyd and Slaughter 2014) |

The longer these concerns are ignored, the more damaging and uncontrollable the process will be. Starting to recognize these concerns will not occur within the institutions that are currently ignoring them. Therefore, alternative processes must be initiated. Firstly, a transformation towards climate resiliency and increasing biodiversity should include differences of every kind and hence should be a global conversation. Secondly, intrinsic development of individuals (intentional) and groups (cultural) deserves more focus over behavioral and social development, as this is often imposed by external institutions. In dealing with global concerns, working on individual intentions is especially largely overlooked and could offer a spectrum of responses within which each person (or group) could find their own truth [86]. This resonates largely with the ways that future thinking (futurology) is described, as a wicked, diversity-emphasizing, skeptical, and futureless concept [88,89] in which people can play around with a diverse set of responses, finding their own truth, grappling with uncertainty and ignorance, and staying skeptical. Moreover, this is a process that strongly resembles the intentional development of individuals and small groups, embracing locality so that tradition and modernity can be connected, to adapt to change and global concerns with respect to traditional culture [90].

This gives us reason to plan and design our cities with a long-term process in mind, allowing for intentional and cultural processes to emerge. The magnitude of both global concerns and these processes requires time, and planning for instant satisfaction is therefore counterproductive. To counteract the immediacy of results, thinking about future generations which we will never meet in person while designing and planning for our society is needed. This cathedral thinking [15] emphasizes the survival of natural and societal systems for multiple (seven) generations.

It also offers a broader perspective on aspects of life we normally find difficult to grasp, such as uncertainty, imagination, complexity, and the paces of changes in our urban environment. In conclusion, the easiest way may be to ignore global concerns and continue the pathway humankind has followed for the last couple of centuries, just like an ostrich who sticks his head in the sand. However, this ignorance of existing complexities would only increase the impacts of global problems.

*4.1. Complexity*

Approaching the global concerns mentioned before requires thinking at multiple levels of complexity. Four levels are distinguished, each dealing with global concerns differently. These levels, pre-conventional, conventional, post-conventional, and integral [91], can be combined with internal/external drivers for change and group vs. individual behavior [92]; a range of ways of dealing with global concerns emerge (Figure 3).

Survival and self-protection dominate the pre-conventional level. Climate change is ignored, and the operating system [93] that fits this is input-oriented and authority-centric. Social standards and passive thinking are related to the conventional level, maintaining the status quo. A dualistic way of thinking is common: climate change is a problem that 'they' will solve. This level focuses on output and efficiency. Transcending rules and embracing reflexivity are common at the post-conventional level and are open to complexity and systemic change to deal with climate change. Society is interested in the outcome and is user-centric. The holistic and integral level includes perspectives across boundaries, embracing appreciative action. Intrinsic values prevail to fit human systems into their natural contexts. This level of complexity is regenerative and ecosystem-centric.

Personal and group development and value levels change over time, with different values becoming more dominant than others. For six of these levels, the way climate change is perceived has been defined (Table 2). The way to deal with global concerns and developing effective climate change responses requires a spectrum of responses, compatible with each of the development stages of both intrinsic and external groups and individuals [92].

When the levels from pre-conventional to integral are interlinked with the value levels, there is a shift from egocentric and authoritarian (red and blue) values, which are dominant

in the pre-conventional level, to systemic and ecological (yellow and turquoise) values, which are more dominant in the integral level (Figure 4).

**Pre-conventional**

| Individual | |
|---|---|
| Has no significance; therefore unable to be comprehended.<br><br>*intentional* | Has no perceivable consequences for action or behaviour.<br><br>*behavioral* |

| Collective | |
|---|---|
| Cannot be seen in the present, so does not require social response<br><br>*cultural* | Infrastructure and provisioning from the past are sufficient<br><br>*social* |

Interior — Exterior

**Conventional**

| Individual | |
|---|---|
| Has happened before, will happen again; fatalism.<br><br>*intentional* | Not my / our problem so no personal action is required.<br><br>*behavioral* |

| Collective | |
|---|---|
| "They" will deal with any issues, if and when they arise; "necessity is the mother of invention"<br><br>*cultural* | Our traditional strengths served in the past and will continue to serve in the future.<br><br>*social* |

Interior — Exterior

**Post-conventional**

| Individual | |
|---|---|
| Climate change alters everything: review values, seek greater understanding; modify behaviour and re-conceptualise needs v wants.<br><br>*intentional* | Personal behaviour change is important and helps drive social change. I can be an example to others.<br><br>*behavioral* |

| Collective | |
|---|---|
| It is imperative to reduce CO2 emissions and reign in consumption, Earth shares and other such social innovations are vital.<br><br>*cultural* | Initiate transformation of infrastructure and reinvention of economics.<br><br>*social* |

Interior — Exterior

**Integral**

| Individual | |
|---|---|
| Accept and value all constructive responses and sympathetically work with others; actively develop new, novel and far-reaching options for self and others; seek broad synergies; all contributions matter.<br><br>*intentional* | An ecology of appreciative actions: accepting, encouraging, respecting, involving, promoting people, actions, strategies and solutions.<br><br>*behavioral* |

| Collective | |
|---|---|
| Develop spectrum of responses across the board and at many levels in a variety of contexts; see cultures as representing different, but partial, answers to the same or similar problems.<br><br>*cultural* | Develop spectrum of practical arrangements, systems, in ways that are appropriate to local needs, cultures etc and reconciled with emergent global imperatives. Nest human systems within natural systems. Growth and development is contextualised in broader spectrum, including the non-material.<br><br>*social* |

Interior — Exterior

**Figure 3.** Typical responses to climate change at different levels of complexity [92].

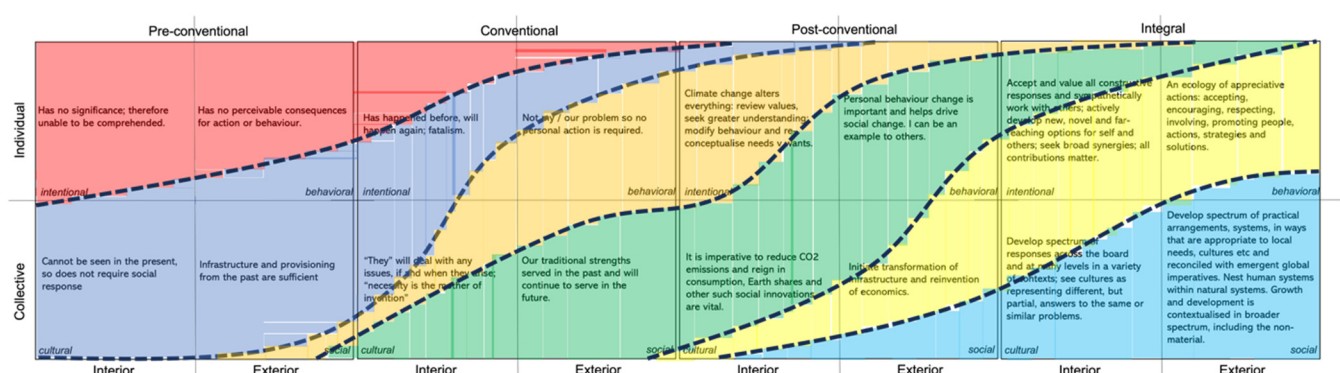

**Figure 4.** Connecting the six development values with the four levels of complexity (by the author, after [92]).

**Table 2.** Value levels and perceived climate change [92].

| Value Level | Perceived Climate Change |
|---|---|
| Red—egocentric and exploitative | It is a jungle out there. Only the strong survive. We must dominate to have a chance. We will do whatever it takes to stay on top. |
| Blue—absolutist and authoritarian | Strong government and a comprehensive set of strictly enforced rules are the only way forward. Overpopulation and pollution will lead to our extinction unless we conform to the imperatives of survival. |
| Orange—multiplistic and strategic | Though times are challenging, we can innovate to survive. There are many opportunities for those who are clever and persistent enough to see beyond the present crisis: new industries, new urban forms, new ways to travel, grow food, etc. Solutions will be found if we step up to the mark and bring the best entrepreneurial intelligence to bear. |
| Green—relativistic and consensual | The communities of the world now have the best reason ever to put aside their differences, work together, and fashion a new world. Joining together in this way, we go beyond the profit motive and seek progress through harmony and love. |
| Yellow—systemic and integral | Chaos is normal—we have been here before. The challenges are great but so is our individual and collective capacity to respond. We can use the crisis to fundamentally shift our civilization to a new level of complexity and systemic awareness. Breakdown leads to breakthrough, and this takes many forms. |
| Turquoise—holistic and ecological | The breakdown of the old world order will be painful but it is a prelude to the new in which cosmic truths prevail, along with principles of "enoughness" and balance at all levels. Individual and community life is centered around, and guided by, holistic and transpersonal energies. Earth resources are no longer limited to the physical and tangible but embrace subtle energies as well. What is meant by the "known universe" expands manyfold. New human and cultural options emerge, and ancient enmities are finally transcended. |

The process of individual and collective growth has an impact on the number of people embracing a certain value set, and changes over time. Where the egocentric and authoritarian groups shrink, the holistic and eco-centric groups grow. This implies that a larger portion of the population will start to embrace regenerative values. Therefore, responses within the integral complexity level will eventually gain larger support, and adaptive capacity in societal and urban systems increases [94–97], just like for ecosystems. In such a systemic view, humans are embedded in natural systems, allowing the adaptive cycle [98] to regenerate, which leads to a higher level of complexity [99], a higher adaptive capacity, and better chances to cope with disturbances such as the global concerns mentioned before.

In conclusion, the way to deal with global concerns is to include multiple responses, allowing processes of individual and collective growth to emerge. This eventually evolves towards higher complexity (integral level of complexity) and regenerative and systemic change as a collective response to global disturbances. The pace of this process, especially in the context of rapid change in cities, requires multiplicity.

### 4.2. Pace of Change

In cities, multiple complexities are interconnected with different paces of change, fast and slow urbanism, and suddenism [100]. The discourse of urban and economic growth often leads to a one-dimensional orientation of outputs and efficiency (connected with the conventional level), and a single, fast pace of urban development, in which building houses and constructing infrastructure are core urban uses (Table 3). A more collective and caring attitude (linked to the post-conventional level) is needed to accommodate urban uses that tend to be more vulnerable and develop slowly, such as nature, water, and food. Besides fast and slow urbanism, sudden change, resulting from disturbances, requires yet another response that is systemic and caters for self-organization of the urban system.

Suddenism [100] is a regenerative urban strategy that allows nature in the city to create a resilient environment that is relatively immune to disasters (and fits with the integral level).

**Table 3.** Three paces of urban change [100].

| Fast Urbanism | Slow Urbanism | Suddenism |
| --- | --- | --- |
| Housing | Ecology and water | Floods |
| Economy | Food/urban agriculture | Bushfires |
| Traffic and parking | Social cohesion | Earthquakes and tsunamis |
| Logistics | Culture | Heavy rain/severe droughts |
| Calculation | Creation | Intuition |

Developing and planning the city should incorporate all three paces of change, and therefore the complexity levels that come with them. Spatially, this means that a very diverse pattern of highly complex, self-organizing, and very adaptable areas; moderately complex and slowly changing places; and less complex but fast-changing sites will spread around the city. Depending on the urban context, all three categories of urban change need to be planned and designed with expert precision.

The complexities and different paces of urban change emphasize the uncertainties of current times. Besides global problems and dynamic urban environments, multiple uncertainties can be dealt with by looking through the lens of multiple futures [101].

*4.3. Uncertainty*

In dealing with uncertainties, the best way is to see multiple 'tomorrows', the extended present, the familiar future, and the unthought future [101] in parallel. Postnormal times are characterized by increasing uncertainties, disruptive events, and shocks. In this context, predicted and predictable futures still form the mainstream of how the future unfolds. For a large part, current trends can be extrapolated (an extended present) or can be constructed and imagined (as familiar futures). A rationalized and controlled response is sufficient to deal with a predictable future. However, when the future is unpredictable, or even unimaginable (Figure 5), such ways of seeing no longer suffice. Uncertainties become too large to manage or control, and the future has to be navigated by extending the ways of seeing the future (unthought futures) [101].

Most policy practices focus on predicted and predictable futures and respond in a mechanistical and calculatable way. A problem is defined and followed by a rationally argued solution. Even in disruptive times, many policies exclude highly uncertain events (at the externalities of the bell curve in Figure 5), because these hardly occur or cannot be imagined anyway [102]. These unpredictable and unimaginable futures are potentially the most disruptive and therefore contain the highest risks. Due to their deeply uncertain nature, a response can never be calculated, and an alternative response should replace standardized solutions. Only by increasing the overall resilience of the natural system, the biggest uncertainties can be dealt with. Nature is capable of self-organizing, hence creating a stable resilient environment that is responsive to any kind of potential disruption.

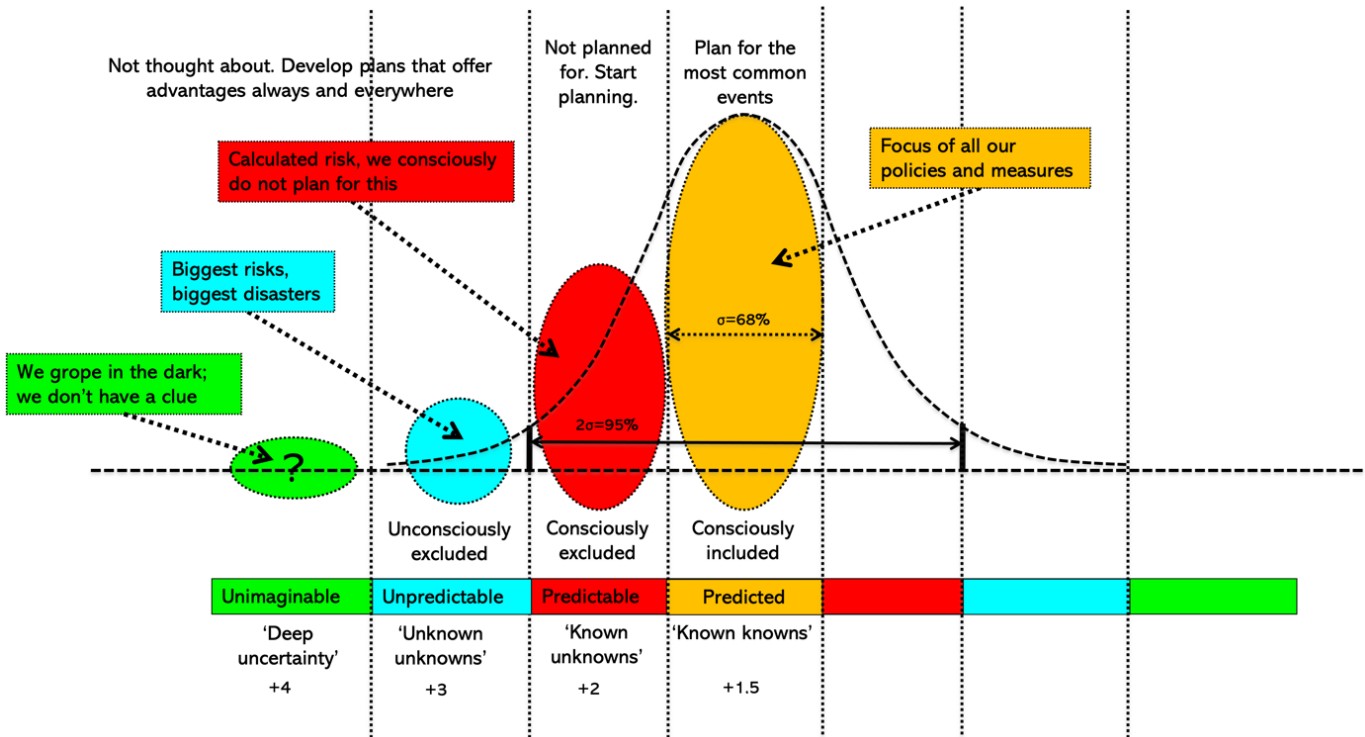

**Figure 5.** Types of uncertainty and the possibilities of response [102].

*4.4. Imagination*

In a context in which complexities grow, urban transformations occur at different paces and uncertainties increase, and solutions need to be explored beyond rationality, logic, and analysis. Therefore, possible and imaginable futures (Figure 6), rather than predictable ones, are more useful. Postnormal times are driven by complexity, chaos, and contradictions, which require enhancing social virtues, individual responsibility and ethics, and imagination [103]. Imagination replaces a probabilistic approach, which responds to well-understood problems in a rational way. Imaginable futures and speculative solutions [104] are needed when conditions are more unpredictable [100].

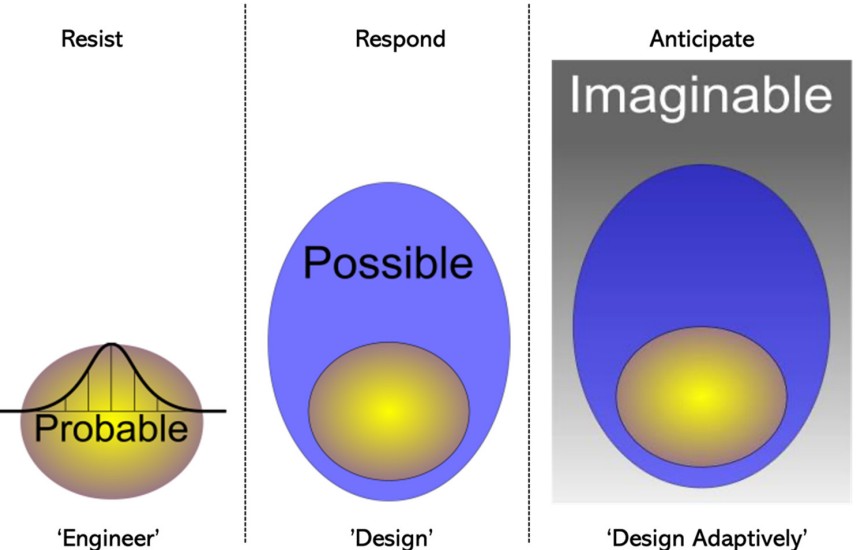

**Figure 6.** Probable, possible, and imaginable [100]; elaborated from [105].

Urban planning and development are hindered by the cultural inability to embrace complexity, transformations, and uncertainty by using an imaginative perspective in the longer term. This has impacted the effectiveness of the many sustainable urban planning concepts and prohibited implementation beyond pilots and experiments.

For this, the mindset of humankind needs to shift towards thinking in terms of long-term futures and being part of nature.

## 5. Eco-Cathedric City

The inability to fundamentally shift towards ecologically founded cities requires a novel concept. In this section, the foundations and pillars for the Eco-cathedric City are presented.

### 5.1. Foundations

Foundations for an urban development that could respond are therefore to be found in the way we see the relationship between ourselves and nature with a long-term perspective on the future.

#### 5.1.1. Long-Term Future

To circumvent the current thinking on short-term actions, survival and (economic) gain, and anticipating uncertainties and complexities, several principles regarding the longer term are useful to incorporate into urban planning and design. The future that current generations establish will impact multiple generations. Building the cities of today will therefore have to benefit all generations that come after the current. This way of cathedral thinking [15] transcends the focus on shortsighted instant satisfaction and gives cities the opportunity to provide the basis and conditions for livability and survival of the urban society for many generations to come. In this context, it is valuable to consider that the future is often not adhering to careful planning, but that a single, totally unforeseen event can shape the future in a direction that cannot be expected [106]. The agility to adjust future visions is not supported by rigidity and short-term gains. Images are important for determining ambition for the far future and stimulating actions in the present to start moving in that direction [107].

#### 5.1.2. Human–Nature Relation

The second foundation lies in the way humankind sees and treats nature. Instead of seeing nature as a resource, perception increasingly dictates not to live *off* nature, which eventually created the Anthropocene [9], but to live *with* nature [108], and human-centric thinking should be replaced by life-centric thinking [109]. In such a view, nature and culture form one cosmology [16]. Humans, creating a beneficial environment for their own lives, are no different than any other species, folding their surroundings to become more favorable for their own development. This is a universal feature of everything that is alive [110,111], denying the need for a separation between human and nature.

This leads to two fundamental observations [16]:

1. Nature and culture are not separable; they form one set of a whole, and all living organisms belong to one system [112].
2. Every organism not only adapts to its surroundings as well as possible to find the right fit to survive [113]; it also modifies and folds the environment for its own personal benefit [110], e.g., to increase its chances to live and survive.

The city is all at the same time: green natural spaces that are isolated within the city, which are large enough to host unique ecologies; it needs green and blue infrastructures that are large enough to establish connections and provide the conditions for a resilient water system; and it needs green space where resources can be generated, harvested, and regrown to provide all that the urban population needs. It therefore needs a contingent nature and green system that is creating the conditions and environment within which urban activities can flourish and be sustained. This means that natural conditions determine

what and where and how urbanity occurs. The relationship between nature and humans has evolved from contrast, contact, and contract to being contingent (Figure 7).

| | *CONTRAST* | *CONTACT* | *CONTRACT* | *CONTINGENT* |
|---|---|---|---|---|
| *Image of nature* | *Wilderness* | *Accessible nature* | *Ecosystem services* | *Indistinguishable* |
| *Formal interaction* | *City and nature have sharp boundaries, protected areas* | *City and nature intertwine* | *City and nature take each other's form* | *Nature directs urban form* |
| | *Bring the city to nature 'satellites and 'garden cities'* | *Insert nature into the city 'green wedges' and 'parks'* | *Go for a complete mix, 'reweaving the urban tapestry' and 'broadacre city'* | *Landscape and ecology first, second and last in designing the city* |
| *Functional interaction* | *City and nature are each other's jungle* | *City and nature come to each other's rescue* | *City and nature take on each other's form* | *City utilizing and mimicking natural processes* |
| | *Places to get lost* | *Regulated leisure in nature* | *Produce food on your own garden lot* | *Live and work within natural boundaries* |
| *Physical interaction* | *City and nature keep their distance* | *City and nature exchange information* | *City and nature take on each other's construction* | *City connecting to its hyperlocal Bioregion* |
| | *Natural expression of the city, 'non-human' outside* | *Natural expression of the city 'well-tempered' environment outside* | *Expression of city and agriculture 'new hybrids' in – and outside the city* | *Ecological processes and rewilding of urban space* |
| *Vision of the city* | *From 'Cabanes' to 'Metropolis'* | *Green-Blue infrastructure to Lobe city'* | *From 'Subtopia' to 'Metabolic City'* | *From 'Nature-Based City' to the City as 'Garden of Eden'* |

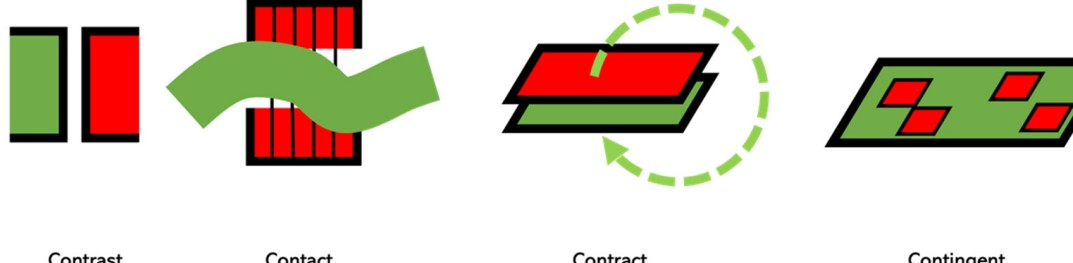

Contrast          Contact          Contract          Contingent

**Figure 7.** Relationships of nature and city [114] elaborated from [115].

5.1.3. Symbiosis

Planning for a truly regenerative city therefore requires rethinking our mental model. Embracing complexity, the paces of urban change, uncertainty and imagination, and the contingency of nature in the human world could enforce a shift in mindset that creates cities which are vital in the long term and can deal with unexpected and unprecedented changes. The perspective slowly shifts from a human-centered to a nature-driven one (Table 4).

**Table 4.** Human-centered and nature-driven perspectives.

| | Human-Centered | Human-Nature Balanced | Nature-Driven |
|---|---|---|---|
| Design strategy | Probable | Possible | Imaginable |
| Type of uncertainty | Predictable | Unpredictable | Deep uncertainty |
| Design approach | Engineering | Design | Adaptive design |
| Urban change | Rapid growth | Shrinkage | Climate impacts |
| Resilience type | Resist | Respond | Anticipate |
| Pace of urban change | Fast urbanism | Slow urbanism | Suddenism |
| Urban problems | Housing<br>Economy<br>Traffic and parking<br>Logistics | Ecology and water<br>Food/urban agriculture<br>Social cohesion<br>Culture | Floods<br>Bushfires<br>Earthquakes and tsunamis<br>Heavy rain/severe droughts |
| Type of complexity | Stable | Complicated | Complex |
| Type of action | Calculate | Create | Intuitive |
| Urban nature relation | Contrast/contact | Contract | Contingent |

In periods of (deep) uncertainty and disruption, nature can build resilience and embrace complexity and intuitive responses to anticipate an unknown future. A good example of allowing the strength of nature to build natural resilience is the Zandmotor (or Sand Engine) as a coastal protection solution along the coastline of the Netherlands [116]. This Building with Nature [117] solution allows the North Sea to shape the coastline in a natural way, directing the sand that is dropped in the sea to self-organize. After a period of several years, it has become clear that dynamic though self-sustaining coastal protection has been established by nature itself [118], meanwhile improving habitat diversification in the marine and terrestrial coastal zone [119]. Humans can then focus on those imaginable solutions that create the space for symbiotic natural processes. This is when a contingent nature, uncertainty, complexity, intuition, and imagination come together at the heart of a new narrative for the city (Figure 8): the Eco-cathedric City. The unease of dealing with uncertainties and complexities is embraced and turned into future urban symbiosis.

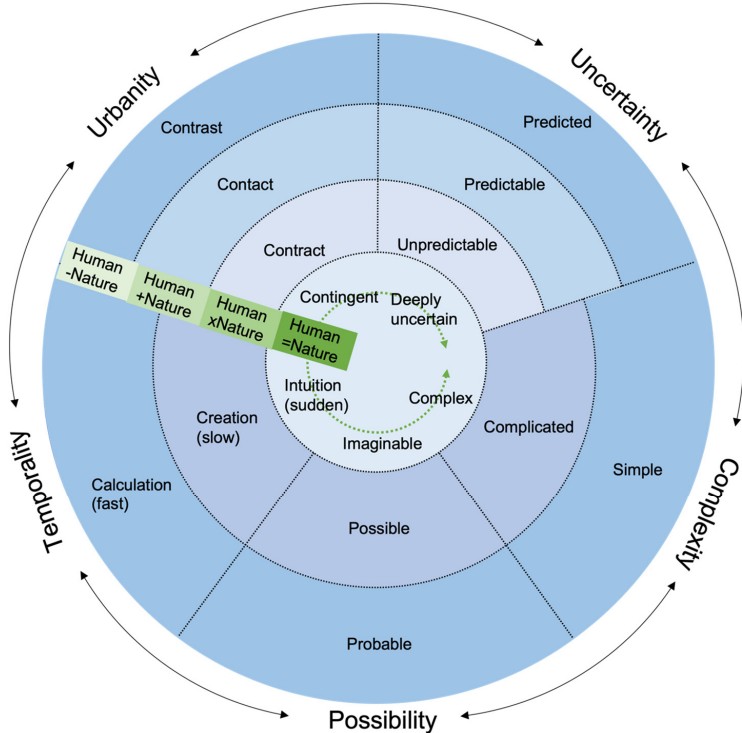

**Figure 8.** Symbiosis of future uneasiness into foundations for the Eco-cathedric City.

### 5.2. Pillars

The Eco-cathedric City builds on the self-organizing power that is common in nature. This fundamental new way of urbanization applies power to three factors: the time perspective of urban change (eco-time), the way the city is designed (eco-space), and how decision making occurs (eco-cracy).

#### 5.2.1. Eco-Time

According to the seventh-generation principle [120], the purpose of realizing a building is not to enjoy it as designers but to provide future generations with the benefits the building holds. The building of literal cathedrals illustrates this approach. Münster cathedral was constructed over centuries [121], and is still being modified, while the construction of the Sagrada Familia in Barcelona was purposely not finished before the life of Antoni Gaudi ended [122,123]. Cities mature over longer periods of time; in ecology, the oak tree takes approximately 30 years before it grows its own acorns [124]; and it takes thousands of years for peatland to accumulate 1.5–2.3 m of deposits [125]. The eco-cathedral which was built by Louis le Roy in the northern part of the Netherlands [126] illustrates the role of nature in the building process. Le Roy defined such a *Gesammtkunstwerk* as a spatial landscape or urban structure, based on mutual participation between humans, plants, and animals which may develop—endlessly in space and time—to a natural climax form of culture and nature [127,128]. The free, creative energy of people is used to build something beautiful together, to surrender it to nature, to build again, to surrender, and so on. It is especially important to give the building system time, as nature develops very slowly. That is why an eco-cathedral has no final realization date. The project will continue as long as nature is allowed to take its course [127]. Moreover, nature practices a very effective way of using, reusing, and returning resources: it is a regenerative system with the aims to sustain and develop its ecosystem over a longer period. Year after year, its processes are directed towards the growth of the ecosystem.

Traditional ecological knowledge emphasizes life in symbiosis with nature [129], intelligently harnessing energy and adapting to environmental obstacles. Survival of the most symbiotic [130] is key, amplifying mutually beneficial interactions. The following levels, each a symbiosis of human and nature, are distinguished [131]:

1. Local knowledge of plants, animals, soil, and landscape as a foundation.
2. Resource management based on local environmental knowledge and ecological processes.
3. Community and social organization for coordinated and collaborative governance.
4. Worldview of general belief systems, interpreting observations of the world around us.

This forms the basis for a long-term vision for the urban landscape of Northern Groningen, the Netherlands (Figure 9). In the Moeder Zernike plan [74,75], the historic natural dynamics of the land-to-sea transition guide land use. Nature and landscape ecology guide agriculture and urban use. Due to the dynamic of the seawater entering the protected hinterland, the level of the land rises along with sea-level rise. Wetlands and newly developed peatlands protect the current towns and urban cores, and allow for saline agricultural practices, such as lobster and oyster farming, saline crops, and fisheries. The implementation takes up to a century or longer and will be altered along the way. However, the guiding principles of the local landscape ecology and dynamical processes will continue guiding the future of this landscape.

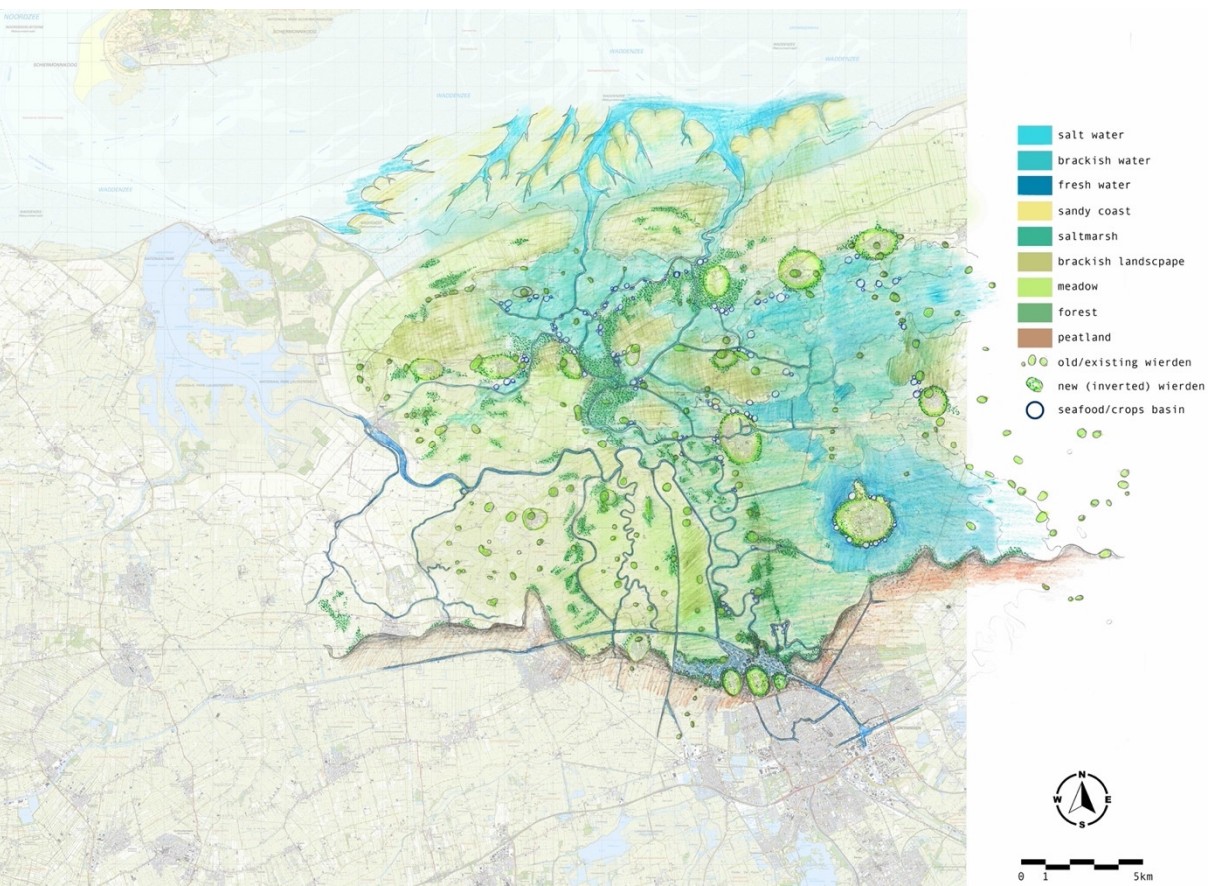

**Figure 9.** Plan for Moeder Zernike [74].

### 5.2.2. Eco-Space

Many plans and designs for urban landscapes focus on the end image, a desired future often represented in a detailed drawing. However, these plans often do not match the future, which turns out differently than expected. The process of change is more important than the final, planned, outcome. Ecosystems evolve and constantly adapt to sudden alterations in the environment. Spatial transformation can be achieved by small and essential interventions to reach higher levels of resilience, as a form of eco-acupuncture [132]. Swarm planning [133] suggests intervening where uncertainty is largest and starting an emergent process of landscape forming that is guided by the forces of nature.

The weakest point in the coastal protection of the Eemsdelta coast is turned into a floodable landscape [134–136] by opening a weak spot in the dike (Figure 10), starting the process of terraforming. This way, the water will slowly enter the hinterland, allowing the landscape and its inhabitants to respond. When conditions change by water entering, floating houses can move up and down with any water level.

### 5.2.3. Eco-Cracy

In human-driven urban planning practice, complex decision making is in general the domain of a select group of privileged people. This often ends in decisions for building projects in vulnerable landscapes or that are unnecessarily put at risk of climate hazards. This caused an increase in flood risk during the 2011 Brisbane floods [137] and significantly increased damage as a result of impervious surfaces and the urban development of wetlands in Florida [138]; the development of coastal cities increased the damage of floods in South Korea [139], and the risk of flooding increased significantly through the narrowing of the riverbed by urban development in Arnhem, the Netherlands [140]. Many of those decisions are not guided by nature's interests per se, which generally tends to make a dazzling amount

of mini-decisions simultaneously. In an eco-cracy, the human eye is harmonized with the eyes of animals and plants [141], emphasizing the interest of non-human species [142] through a way of nature 'doing non-doing' (negatively restraining evil things for life), and embracing spontaneity and intentionality [143]. Ecocracy as a governance system [144] reflects the principles of deep ecology: the idea that humans and the environment are of equal value and for that reason humans have no right to dominate nature [145]. Making decisions about the urban landscape therefore requires a broader variety of participants. If the complexity of life in those landscapes is represented in the complexity of the decisions taken, one of the questions to ask would be how the environment would look if nature were to decide. This could be achieved, for instance, by appointing nature as the CEO of a company [146,147] or by giving natural entities such as the Te Urewara River in New Zealand [148] or the Wadden Sea, the Netherlands [149], (juridical) rights. The key point here is how communication with 'nature' is shaped. Because nature does not speak as humans in boardrooms and courthouses do, an alternative language is required. In the city of Monterrey, Mexico, initial experimental representations of nature in the decision-making process are planned, in which certain elements of nature, such as the mountain, the river, the forest, and/or keystone animal or plant species, take a seat at the conversation table. Although these elements will be represented by human beings, these people will express the way decisions influence the being, feelings, and ability to (keep) functioning, for this specific relevant area, of elements of nature. In the discussion, their opinions are evenly valued as traditional considerations about economics, infrastructure, or housing. The expectation is that by integration of arguments that are important to natural elements, the decision making will be different.

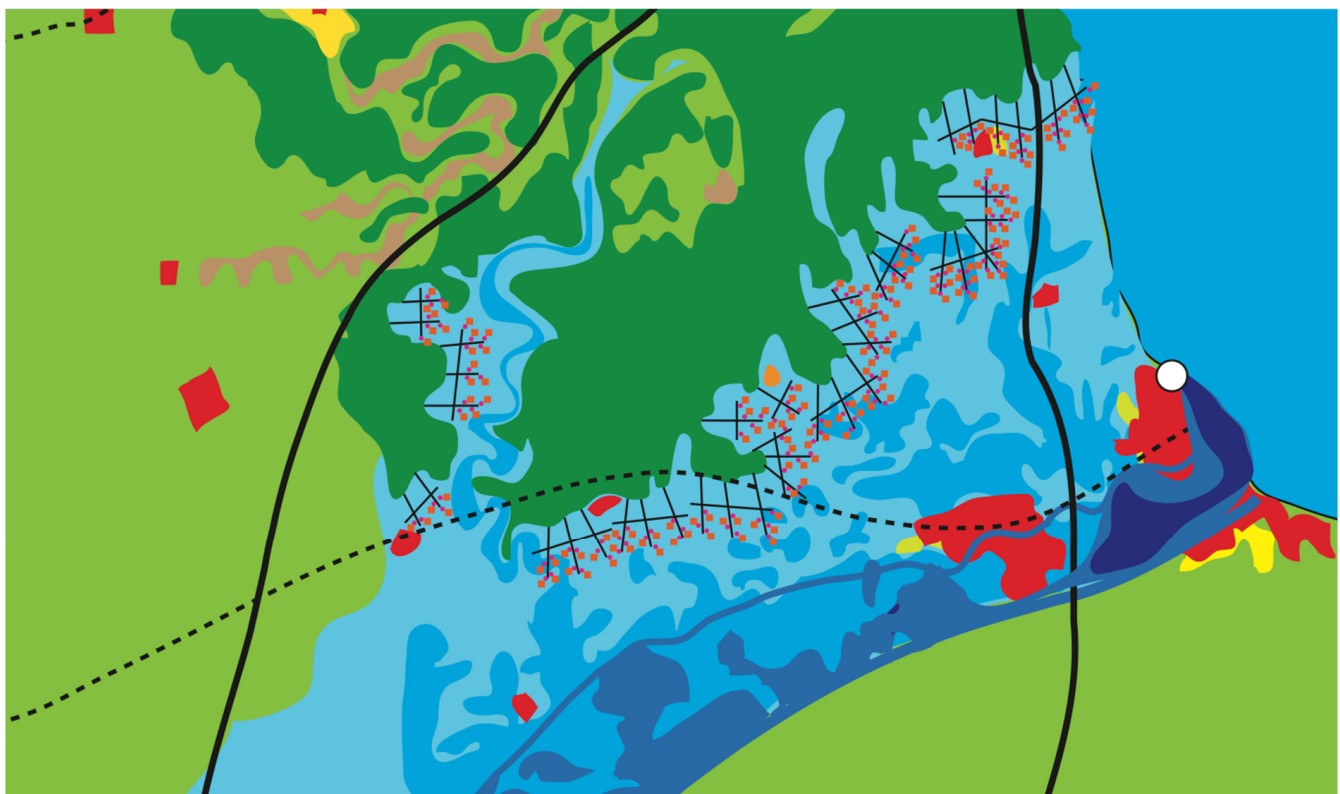

**Figure 10.** Floodable landscape, Eemsdelta [100].

## 6. Discussion

In the era of the Anthropocene, the urban society dominates and uses nature and natural resources for its own benefits and comfort. To date, nearly all urban landscapes are still designed from a human-centric viewpoint. However, given the current polycrisis [150],

continuing the pathway of human dominance does not seem to lead to sustaining human life on Earth. The paradox here is that the more humans try to protect humanity, the more nature is exploited, and the more human life is in danger. The proposal for designing Eco-cathedric Cities is only the start of the transformation process, as in practice, the planning and design community has not yet discovered the challenge of what a rethink of the relationship between nature and humans might encompass.

The question is how to accelerate the implementation of such a new paradigm. Actuality and day-to-day decisions face difficulties with abstract concepts such as multi-generational thinking, non-finite ways of planning, and allowing non-humans a place in governance.

Therefore, further research needs to be conducted:

- How can nature, natural entities, or single species obtain a prominent role in decision making?
- How can the 'seventh generation' become part of choices made today?
- How can small, crucial spatial interventions lead to the desired change towards resilience?

These questions are not yet part of spatial and/or governance science nor tested in practice. The Eco-cathedric City shall therefore be investigated in these three directions, symbiotically, to gain evidence of whether, how, and where nature guides an urban landscape in which all living organisms are sustained.

## 7. Conclusions

This article proposes an alternative way of thinking about urban landscapes. The Eco-cathedric City emphasizes a new balance between the natural and human. The way nature can self-organize promises novel avenues towards designing for invulnerability of the urban landscape.

Three aspects of the Eco-cathedric City are seen in coherence:

- To take a much longer timeframe than the current planning paradigm is used to.
- To embrace a planning approach that is grafted on small and crucial interventions that enforce change of the urban system in the desired direction.
- To involve underprivileged humans, other organisms, and natural entities in spatial decisions, of which there are good examples available, such as social urbanism practiced in the city of Medellin, Colombia [151–153] or the development of the FoodRoofRio, a productive roof which was codesigned and build by and with the local residents of the Cantagalo Favela in Rio de Janeiro [154]. This is a long-term investment in building trust amongst the residents, exploring commonly felt problems, and developing solutions that are supported by all. A good example of this process has been taking place since 2014 in two disadvantaged neighborhoods of Campana-Altamira in Monterrey, Mexico [155].

It is recommended to adjust the way urban landscapes are planned for:

- Start with understanding the foundations of the local ecosystems: its plants, animals, the soil, and the landscape.
- Use this knowledge to design a self-organizing ecosystem in which regenerative resource management is prioritized. Such a resilient ecosystem is paramount for further urban development and therefore is given the spatial conditions to function in its most natural way.
- Create social constructs to support the emergence of resilient ecosystems and subsequently determine where and how human uses fit within these boundaries.
- Harness an evolving belief system with core values such as reciprocity and symbiocity.

In current times, the dominant rule of man over nature is dysfunctional, even dangerous. In a nature-driven urbanization process, Eco-cathedric Cities give nature a central position in (urban) decisions taken. Humans need to reroot their relationship with nature from superior to symbiotic [129].

**Funding:** This research received no external funding.

**Data Availability Statement:** Not applicable.

**Conflicts of Interest:** The author declares no conflict of interest.

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
