# Peer review of "The Eco-Cathedric City: Rethinking the Human–Nature Relation in Urbanism"

_land, doi:10.3390/land12081501_

Round 1

Reviewer 1 Report

A wonderfully framed work describing the challenges and longer termed urban solutions in an innovatively graphical and coherent manner. My comments range from the very minor (typo, nit-picking level) to ones that propose further elaboration on the points made. Primarily through illustrated case studies of urban eco-interventions, which support the arguments even at a small urban, or even building/facade scale. The authors own design case study (Floodable landscape Eemsdelta 2015) is highly appropriate, but appears quite isolated and I feel needs support from various diverse and realised case studies.

Minor point (Typos):

·       Figure 1. far left image, Planetary boundaries text unreadable when zoomed in.

·       Use of the word ‘of’, perhaps Live ‘off’ Nature? Instead of 358 seeing nature as a resource, the perception increasingly voices not to live of nature, which 359 eventually created the Anthropocene [9], but with nature [108] and human centric thinking 360 should be replaced by life centric [109].

·       Denying 364 the need for a separation between huma (human?) and nature.

More elaboration suggested:

·       nature can build resilience, embraces 391 complexity and intuitive responses to anticipate an unknown future. A continuing theme, and agree, but can this be evidenced? Case studies study would help here. References and brief sentences of outcomes would suffice.

·       Examples/case studies of urban change (eco-time), the way the city is designed (eco-space) and how decision-403 making occurs (eco-cracy). Brief summaries and illustrations would be beneficial.

·       In the human-driven urban planning practice complex decision-making is in general 453 the domain of a select group of privileged people. Decisions are not guided by nature´s 454 interests per se, which generally tend to take a dazzling amount of mini-decisions simul-455 taneously.  Again, any examples to put forward that show those failings in all their apocalyptic glory?

·       In the eco-cracy described, how does communication with nature occur? What language and questions? How does/can nature potentially negotiate with humans? Can examples at any scale be included, no matter how small. Perhaps realised smaller inventions could be reinterpreted and supersized to support your conclusions?

·       Very interesting and significant to read about the ‘seventh generation’ and how this should influence urban decisions. Could this ‘seventh generation’ be depicted in a graphical timeline alongside an imagined future eco-cathedral development? Perhaps there are realised projects that have and are partly way through those their generational timeline of development?

·       To involve underprivileged humans and other organisms and natural entities in 498 spatial decisions. Any examples to enforce this, perhaps from the Just city masterclass speakers might help here? Particularly Alejandro Echeverri’s realised work with communities?

Hope this helps, and I look forward to reading the finished publication!

no comment

Reviewer 2 Report

This article presents an alternative approach to urban planning ('planning of urban landscapes' in author’s own words, which in my opinion is not totally accurate, since the base of urban planning is the planning and the configuration of uses, not the physical design of space). However, as the author points out in the discussion section, all the elements that will allow the application of this approach in everyday planning practice are still to be worked out. This reminds me of the case of post-modern planning approach, which had a well-developed theoretical background that was impossible to be translated into practice. In my opinion, theories in applied sciences (such as urban planning) need to go all the way down to practice. I know that this is a goal impossible to achieve in a single publication. However, I also believe that it will be to the benefit of this article to include some ideas on what the ecocathedric city approach entails in real-life planning practice. Also, it will be useful to the reader if the author could put his approach within the existing landscape of the planning theories and approaches.   

Round 2

Reviewer 2 Report

I feel that my comments didnt find their way into your manuscript, but I understand that there are different schools in urban planning and obviously we dont belong to the same. I also believe that different approaches should be presented and published, especially for such a nodal issue as the one that this paper deals, thus I totally support the publishing of this -otherwise very well written- manuscript. I wish, however, one of your future publications to specifically focus on the implementation of the eco-catherdal approach, especially since the examples you refer in the manuscript (Zandmotor, Sagrada Familia, Medellin, FoodRoof ...) do not comprise typical examples of urban planning interventions. I also believe that this is a paper with the potential of positive influence, thus I warmly support its publication.